# A Systematic Review of Risk Assessment Associated with Jellyfish Consumption as a Potential Novel Food

**DOI:** 10.3390/foods9070935

**Published:** 2020-07-15

**Authors:** Guglielmo Bonaccorsi, Giuseppe Garamella, Giuseppe Cavallo, Chiara Lorini

**Affiliations:** 1Department of Health Science, University of Florence, viale GB Morgagni 48, 50134 Florence, Italy; guglielmo.bonaccorsi@unifi.it (G.B.); giuseppe.cavallo@unifi.it (G.C.); chiara.lorini@unifi.it (C.L.); 2School of Specialization in Hygiene and Preventive Medicine, University of Florence, viale GB Morgagni 48, 50134 Florence, Italy

**Keywords:** novel food, food safety, jellyfish, risk assessment

## Abstract

FAO (Food and Agriculture Organization of the United Nations) predicted that the world’s population will reach over 9 billion in 2050. This condition will require an increase of the global food production by 60%. Technology and scientific research in the near future will soon be oriented towards optimizing the limited existing resources, reducing waste, and improving the consumption of sustainable new foods. Jellyfish could be a valid alternative among novel food. The purpose of this systematic review was to assess microbiological, chemical, physical, and allergenic risks associated with jellyfish consumption. Four research strings have been used to assess evidences about these risks. PRISMA (Preferred Reporting Item for Systematic Reviews and Meta-analysis) guidelines were applied. Finally, 14 articles were found. Results showed a good level of health safety for jellyfish consumption in terms of its allergenic and microbiological risks. No evidence was found about physical risks. As regards chemical safety, it should be fundamental to carry out a constant monitoring of the water where jellyfish are captured or bred. Periodic checks will be necessary on the finished product, such as the analysis of the aluminum content commonly used during the manufacturing process. The number of publications found was rather small, and further investigation will be necessary to enforce the knowledge on jellyfish consumption by humans.

## 1. Introduction

The last 100 years have seen a significant growth of the world population, estimating that it has increased from 1 billion people in 1800, to 2 billion in the first three decades of the twentieth century, to 7.8 billion in 2020. The world population growth rate peaked in the early 1960s with an annual increase of around 2%, higher than the current one which is about 1.05% [1]. Although the growth rate is gradually decreasing, the Food and Agriculture Organization of the United Nations (FAO) estimates that the total population will reach 9 billion people by the mid-2050s [2].

Since the 1960s, with the Green Revolution (GR), food production increased to respond to the needs of the growing population worldwide [3]. Thanks to the widespread adoption of GR approach, large-scale famines and social and economic upheavals were averted. The technological advance that led to the dramatic achievements in world food production over the last decades was the development of high-yielding varieties of cereals, in particular those varieties of wheat and rice [4]. On the other hand, the GR has brought several negative impacts on the environment, which poses a threat to the sustainability of agriculture and natural resources [5].

In fact, agricultural practices that use high amounts of external inputs, such as inorganic fertilizers, pesticides, and other amendments, can overcome specific soil constraints to crop production: in the most intensively managed systems, this has resulted in environmental degradation of soil, vegetation and water resources [6]. For the near future, the challenge, declared in Agenda 2030 of the Sustainable Developments Goals (SDGs), is to provide food to everyone with a sustainable production system [7].

Owing to the world population growth, it is estimated that global food production will have to increase by 70% in 2050 [8]. The increased demand for animal-based protein is expected to intensify pressure on land due to the need to produce more animal feed [9]. Serious environmental problems, related to food production and consumption, include climate change, clearing forests, water pollution, water scarcity, soil degradation, eutrophication of water bodies, and loss of habitats and biodiversity. Food consumption is associated with the bulk of global water use and is responsible for the generation of approximately one-fifth of greenhouse gas emissions (GHGs) [10]. It is predicted that some aquatic species, like jellyfish, will grow as global temperatures continue to rise [11]. In fact, recently, jellyfish numbers have exploded in the Mediterranean due to the result of various factors including oxygen depletion; increase of plankton by eutrophication; depleted populations of large predators such as red tuna, swordfish, and sea turtles that feed on jellyfish; temperature changes; and water contamination. Pollution, too, may be fueling the jellyfish explosion.

Jellyfish may, in fact, be one of the few foods that we could remove from the sea that would have a net positive effect. They are an invasive species, known for invading beaches and raiding fish farms [11,12].

The methods of food production in the near future, thanks to the aid of technology and scientific research, will have to consider the reduction of waste, the earth’s resources, the environmental sustainability, and consequently the chance to consume new foods to respond and satisfy this growing demand [13,14]. Among these food choices, jellyfish can be a valid alternative to conventional proteins with low carbohydrates and saturated fat intakes; moreover, their consumption could also aid to avoid the gradual loss of fish biodiversity [12]. Undernutrition and malnutrition exist to a considerable degree in both industrialized countries and countries in transition. Even in Europe, about 5% of the overall population is at risk of malnutrition, and among vulnerable groups—the poor, the elderly, and the sick—this percentage is still higher. At the same time, people worldwide face an increase in such food-related health problems like cardiovascular disease, obesity, and diabetes because of rich foods, modern diets, sedentary lifestyles, and overeating. Key diet-related factors are the high intake of saturated fat, salt, and sugar and the low consumption of fruits and vegetables [10]. In this context, jellyfish could have an important role as good source of quality nutrients to adopt in Western diet as well as in developing and underdeveloped countries, particularly for protein content. All jellyfish, in fact, tend to have low caloric values (1.0–4.9 kcal/g D.W.) combined with negligible fat contents (0.4–1.8 g/100 g of Dry Weight, D.W.), while protein (20.0–53.9 g/100 g D.W.) and minerals (15.9–57.2 g/100 g D.W.) are the richest components. Total collagen content of edible jellyfish varies from 123 to 694 mg/g D.W. [15]. It was estimated that there are 200 species of jellyfish (*Scyphozoans*) that are divided into four orders: *Semaeostomeae*, *Coronatae*, *Rhizostomeae*, and *Stauromedusae* [16] of which only the *Rhizostomae* in the mature stage are considered edible [17].

Anatomically, the main parts of a medusa are umbrella, tentacles and oral arms (covered with cnidocytes and venom). Jellyfish stings may induce immediate or delayed toxic reactions. Upon contact with the tentacles, the cnidocytes may penetrate the skin and inject the venom, causing nematocyst dermatitis, anaphylactoid reaction, shock, and even death [18]. The edible part is the umbrella, while the stinging tentacles, the reproductive and digestive tracts are removed [17]. Studies demonstrated that the umbrella of this species contains relevant concentrations of long chain fatty acids, antioxidants, collagen, and taurine, making it a possible source of food supplements [19]. In *A. hardenbergi*, *R. esculentum*, and *R. hispidum,* very low caloric values were found. Comparatively, jellyfish oral arms were found to have significantly higher energy density than the bells. The distribution of protein and ash content of jellyfish differed significantly in different body parts irrespective of species. It was found that water comprised 95–98 g/100 g wet weight (W.W.) of the jellyfish studied, and a hygroscopic 12–20% g/100 g of their D.W. Jellyfish are reported to contain very low amount of cholesterol. All edible jellyfish considered in this study were found to contain very low level of calories but high in proteins [15]. *Aurelia* sp. (commonly known as moon jellyfish), *Cotylorhiza tuberculata* (known as fried-egg jellyfish), and *Rhizostoma pulmo* (known as sea lung jellyfish) from Mediterranean coastal areas were investigated by Leone et al. [20]. The analysis revealed that none of the jellyfish protein samples contained the essential amino acid (EAA) tryptophan (Try). All the remaining EAA were found in *R. pulmo* and *C. tuberculata* specimens. The ω–3 PUFAs were abundant in the three jellyfish species, with the ratio of ω–6 to ω–3 resulting always in favoring of ω–3 fatty acids.

In a study conducted by Raposo et al. [17], sensory properties of jellyfish are also described. The texture was characterized as cohesive and firm, moderately juicy, hydrated, and having low adhesiveness, which allows adequate chewing. A moderate or strong salty taste or flavor was perceived by a sensory panel that referred a slightly or moderately salty aftertaste. Both salty taste and salty aftertaste may be associated with sodium chloride content. The main impact of the product on the mouth was defined as freshness. Pedersen et al. [21] note that jellyfish is “a food material mostly uncommon to the Western palate, but a delicacy in traditional Asian cuisine having a gastronomic history of more than a thousand years. It is eaten mainly for its interesting crunchy mouthfeel resulting from a monthlong salt preservation using sodium chloride and alum. This preservation drastically changes the texture of the jellyfish from being gel-like to resembling that of pickled cucumbers”.

Fishery, markets and consumption of jellyfish are currently limited in western countries by the lack of market demand for jellyfish products and the absence of processing technologies adequate to the western market safety standards. Jellyfish as food is not familiar to most of western consumers, and it is negatively perceived. Therefore, new opportunities for expanding jellyfish food uses in western countries will rely on the development of new processing technologies and of market demand, which in turn will depend on increased knowledge of jellyfish as raw food material and on public perception of it [22].

Jellyfish are marketed in the form of products in brine, chilled, frozen, live, or desalted products directly ready for consumption (ready to eat). The technique of processing jellyfish is essentially based on a multi-phase drying process, using mixtures of salt and alum, and jellyfish products are typically consumed after desalting and rehydration in salads or after cooking [22].

Jellyfish is a very ancient food widely consumed in traditional Asian cuisine. It is usually eaten and marketed in Asian countries like China, Japan, Thailand, and Malaysia [17]. In 2001, the annual global harvest of edible jellyfish was estimated to be around 321,000 tons [23]. In 2018, the total marine capture production of *Rhopilema* spp. and *Cannonball jellyfish* was, overall, around 300,000 [24]. Chinese popular culture attributes health benefits to eating jellyfish. In fact, it is believed to be an effective cure for arthritis, back pain, hypertension, and ulcers, producing other benefits like softening skin and improving digestion. Jellyfish is also alleged to remedy fatigue and exhaustion, stimulate blood flow during the menstrual cycle, and ease any type of swelling [25]. Laboratory researches have shown that edible jellyfish are made up of a collagen type that can be enzymatically hydrolyzed to some characteristic peptides with an antihypertensive effect [26]. However, most of these claims regarding the medical value of jellyfish are described in non-scientific publications in Chinese [25]. Recently, the new European regulation on novel food (No. 2015/2283) together with the biochemical characterization and identification of bioactive properties of some Mediterranean jellyfish species paved the way to the potential use of western jellyfish as food, including the latter in the list of the so called "novel food". Before commercializing and consuming these products on a large scale in Europe, however, it is advisable to evaluate the evidences present in literature regarding their safety, i.e., the potential microbial, chemical, physical, and allergenic risks related to human consumption [27]. The purpose of this systematic review is to summarize the state of the art with regard to the potential risks associated with the consumption of jellyfish.

## 2. Materials and Methods 

The Preferred Reporting Item for Systematic Reviews and Meta-analysis (PRISMA) statement was adopted for this work. Four databases were explored by means of ad hoc search strings: Medline, Embase, Science Direct and Web of Science; the timeframe considered is 1 January 2014 to 1 May 2019. A search string was used for each type of risk (microbiological, chemical, physical, and allergenic):For microbiological risk: food and (jellyfish* or “novel food*”) and (microbiota or “microb* community” or “microb* count*” or “microb* load” or “microb* risk” or “microb* hazard” or “microb* saf*” or "food safety”);For chemical risk: food and (jellyfish* or “novel food*”) and (“chemical risk*” or “chemical hazard*” or “chemical safety” or metal* or arsenic or cadmium or copper or zinc or chrome or lead or aluminium or mercury or toxin*);For physical risk: food and (jellyfish* or “novel food*”) and (“physical risk*” or “physical hazard*” or “physical safety” or “foreign bod*”) and radionuclide;For allergenic risk: food and (jellyfish* or “novel food*”) and (allergen* or allerg*).

Only articles written in English, French, Spanish and German were included. For this reason, two articles were excluded due to the linguistic restriction (they were written in Japanese). All duplicates were excluded from the obtained results. For each of the four investigated risks, a flow chart was produced to synthetize the obtained results.

## 3. Results

A total of 1978 results were obtained: 544 for microbiological risk, 456 for chemical risk, 386 for physical risk, and 592 for allergenic risk. The screening by title, abstract, and full text was made, and after removing duplicates between the four databases, 14 articles were selected as the sum of the results of all the search strings. Of these 14 selected works, seven were laboratory research studies, four were case reports, two were trials, and one was a descriptive study. An article entitled “Eating jellyfish: safety, chemical, and sensory properties” was found three times using search strings for microbiological, allergenic, and chemical risk [17].

Analyzing the countries where the studies were made, the publication concerning microbiological risk was conducted in Portugal; for the allergenic risk, three studies were conducted in Japan, one in China and two in Portugal. As for the articles concerning chemical risk, three articles were conducted in China, one in Portugal, one in Spain, one in Australia, and one in Malaysia. No articles were found for physical risk. Flow charts related to microbiological, allergenic, and chemical risks are shown in Figure 1, Figure 2 and Figure 3, respectively.

### 3.1. Microbiological Risk

One study was included in the final synthesis.

About microbiological risk results, no pathogenic microorganisms were revealed. In the research article by Raposo et al. [17], the microbiological profile of a certain jellyfish type, *Catostylus tagi* (*C. tagi*), has been examined; the analysis focused on the research of *Aeromonas hydrophila*, *Listeria monocytogenes*, *Salmonella* spp., and *Vibrio* spp., according to the European Commission Regulation n°2073/2005 and 1441/2007 on food safety. The results showed absence of microorganisms in all the tested pathogenetic markers. No evidence was found regarding contamination with viruses and fungal biota.

### 3.2. Chemical Risk

Seven studies were included in the final synthesis.

According to Khong et al. [15], hazardous elements such as lead (Pb), cadmium (Cd), mercury (Hg), and inorganic tin (Sn) were not within the detection limit (0.01 weight%). Elements in jellyfish, especially trace elements, were highly affected by the habitats of the jellyfish population.

Other findings showed that jellyfish are particularly sensitive to marine pollutants due to the phenomenon of bioaccumulation. According to a study conducted by Epstein et al. that has examined the rate of uptake and retention of trace metals in a jellyfish species (*Cassiopea maremetens*), metal accumulation in jellyfish tissue began rapidly within 24 h of exposure to treated water. Cu concentrations were significantly higher under high nutrient conditions (ANOVA: F1,16 = 7.436, *p* = 0.015), reaching 2.627 ± 0.031 µg/g, an increase of approximately 18.1% from ambient concentrations [28].

Another research conducted by Muñoz-Vera et al. [29] assessed the possibility from a jellyfish species (*Rhizostoma pulmo*) to bioaccumulate trace elements in a Mediterranean coastal lagoon from South Est Spain. This study has evaluated the concentrations of different analyzed elements (Al, Ti, Cr, Mn, Fe, Ni, Cu, Zn, As, Cd, Sn, and Pb) in 57 samples taken by this area. Although the concentrations of these elements were moderate, bioconcentration levels in relation to seawater metal concentration were extremely high. Arsenic concentration was significantly higher in oral arm tissues than in bell tissues at all locations, with values up to four and two times higher in oral arms, respectively. Mean concentrations of iron, zinc, arsenic, manganese, and titanium were higher in oral arm tissue than bell tissue at all locations, with significant differences for different sampling areas. On the contrary, no significant distribution patterns were found for accumulation of nickel, copper, and tin in jellyfish tissues.

Raposo et al. study [17] provided elemental and compound contents in the total solids (D.W.) of the umbrella product and compared it with the raw umbrella of *C. tagi* (Lisbon, Portugal). Of the 25 elements analyzed in the umbrella product, 11 (Al, B, C, Fe, H, K, Mg, Mn, N, Na, and P) showed a significant mass change, *p* < 0.1, compared to the total solids of the raw umbrella. Eight elements (Ca, Cd, Cr, Cu, I, Ni, S, and Zn) showed no significant differences. Particularly, Al was lower in cooked product than in raw; As, Co, Hg, Mo, Pb, Se, and V were not detected after cooking.

Other three works have been conducted in three Chinese cities and they have analyzed aluminum contents in jellyfish that are commonly consumed by people [30,31,32]. Ma et al. [30] have conducted a 6-year study (between 2010 and 2015) to assess the risk of dietary aluminum exposure in residents of Tianjin metropolis. Totally, 21.14% of food samples exceeded the recommended aluminum residue limit over the study period (100 mg/Kg). The lowest mean aluminum levels in food were detected in 2010, and the highest levels were found in 2015. The highest levels were found in jellyfish (433.28 ± 402.11 mg/kg), while the lowest aluminum levels were found in the other aquatic animal food products (2.26 ± 5.58 mg/kg). This was probably associated with its manufacturing process despite the implementation of a new policy on the use of aluminum food additives in this town.

According to the article written by Zhang et al. [31], the average dietary exposure to aluminum was 1.15 mg/kg bw/week (body weight/week), which is below the provisional tolerable weekly intake of 2 mg/kg bw/week. However, the study states that jellyfish was the main Al contributor, providing 37.6% of the daily intake and a mean of 4862 mg every kilogram of product.

The aim of a study conducted by Yang et al. [32] was to evaluate the dietary aluminum intake level in residents of Shenzhen, China. Diets of a total of 853 persons were investigated in three days of food records. One-thousand three-hundred ninety-nine food samples were collected from market to test aluminum concentration. Among them, high aluminum levels were found in jellyfish (ranging from 318.3 to 1000.4 mg/kg with a median of 527.5 mg/kg), and the highest aluminum intake concerned children with an exposure level of 3.356 mg/kg bw/week and 3.248 mg/kg bw/week in 0–2 and 3–13 age groups, respectively, which is higher than the allowed threshold.

### 3.3. Physical Risk

Using a search string on physical risk, no studies were included in the final synthesis because they did not respect the search criteria after selection for title, abstract, and full text according to PRISMA statement guidelines.

### 3.4. Allergenic Risk

Six studies have been included in the final synthesis that refer to this risk.

A case report by Li et al. [18] showed a 26-year-old Chinese male that developed symptoms of erythema, pruritus, and palpitation half an hour after eating cooked salt-preserved jellyfish. He exhibited dizziness, dyspnea, and tachycardia. The patient was otherwise healthy, had no history of allergies to drugs or other substances, and had been stung severely by jellyfish approximately 6 months prior. Within 15 min of eating the jellyfish, the patient began to develop symptoms. After treatment and fluid infusion, the patient was given an oral antiallergic agent (loratadine tablets, 10 mg/d for 1 week) and educated regarding a safe diet. Five days later, the patient’s urticaria had dissipated.

In the study conducted by Amaral et al. [19], 20 subjects with severe seafood allergy and 5 atopic, non-food allergic controls were enrolled. Skin prick-to-prick tests (SPPT) with raw and boiled umbrella were performed, as well as challenges with *C. tagi* umbrella in all subjects. All 20 patients presented negative reactions to the SPPT with the raw umbrella of *C. tagi*. None of the 20-seafood severely allergic patients nor the control subjects had immediate or late phase reactions to any of the pastes. All this confirms the lack of cross reactivity between mollusks, crustaceans, cephalopods, fish, and jellyfish.

In a case report [33], a 45-year-old man presented two episodes of anaphylactic reaction 2 h after dinners including jellyfish salad, accompanied by dyspnea, chest tightness, abdominal cramps, palpitations, vomiting, dizziness, headache, and loss of consciousness. The onset of the nattō allergy was 8 years prior to the episodes considered, and poly–γ–glutamic acid (PGA) was identified as the causative allergen. The patient was a surfer and had been frequently stung by jellyfish. Patient also might have been sensitized to jellyfish nematocyte PGA by stings via the skin during surfing and consequently developed the anaphylactic reactions to ingested jellyfish and nattō. In another study [34], it was reported a case of a 14 year-old boy that developed a cough, urticaria, and dyspnea 30 min after he ate a breakfast that included dried and salted jellyfish, which he rarely consumed. There was no past medical history other than a house-dust mite allergy. He had never been to the sea and had no previous episode of jellyfish contact or sting. The patient presented tachycardia, hypotension, edema, wheezing, and diffuse urticarial lesions. A prick-to-prick test for dried and salted jellyfish was performed to identify the causative food. The patient was diagnosed with anaphylactic shock as a result of jellyfish ingestion. Wakiguchi et al. [35] reported a similar case in a 14 years old boy that developed wheezing and dyspnea 1 h after dinner, which included salted jellyfish. The diagnosis was anaphylaxis following jellyfish ingestion, without PGA sensitization.

The summary of all results is shown in Table 1.

## 4. Discussion

The recent regulatory upgrade of the EU rule on novel foods, the globalization of food markets, and the increased availability of jellyfish local biomass have enhanced the attention on the jellyfish species that colonize European coasts. The latest studies have reported that some native jellyfish species present in the Mediterranean Sea have biochemical and textural features similar to edible Asiatic ones [17,36] and seem to be good candidates as a new “local seafood product”. Jellyfish are extremely perishable raw materials and are generally treated within a few hours of collection to avoid deterioration, thus maintaining the organoleptic and safety qualities [37]. An essential prerequisite for the authorization of a novel food and its inclusion in the Union list by the EU Commission is that it must not cause any safety risks to human health on the basis of the available scientific data [27].

As concerns the microbiological risk assessed in this review, the current state of the art shows that jellyfish cannot represent a source of severe microbiological hazards for humans. Future studies should consider metagenomics and metabolomics approaches for the analysis of raw and processed jellyfish. This could be interesting for obtaining information on total microbiota associated to jellyfish and qualitative and quantitative data on microbial metabolites to estimate other sources of microbiological risk for humans [37]. The allergenic risk could be related to the transformation degree of the product (raw or cooked) and to the length of jellyfish peptides acting as antigens. The studies included in the final synthesis show that patients with allergies to crustaceans, cephalopods, mollusks, seafood and fish can consume jellyfish without increasing risk of an allergic reaction [17,19]. Three case reports of anaphylaxis have occurred after eating raw jellyfish [33,34,35], while one article written by Li et al. [18] describes an episode of anaphylaxis after eating jellyfish that were not only salt-preserved but also cooked. Moreover, it would be appropriate to know if, in the presented case report, the jellyfish have been prepared correctly, eliminating all the non-edible parts like the stinging tentacles and the reproductive and digestive tracts.

Another important finding that has emerged from the analysis of the publications concerns the lack of cross-reactivity among subjects allergic to crustaceans, mollusks, fish, and seafood and the consumption of jellyfish [19].

The results of our research highlighted the probable etiological role of several allergens as a possible factor responsible for anaphylaxis in subjects who ingested a meal that contained jellyfish as an ingredient. Among the allergens, PGA would seem to be one of those antigens implicated in the genesis of anaphylactic reactions. Inomata et al. [33] report the case of a 45 years old man who went through two episodes of anaphylaxis two hours after ingestion of a jellyfish salad. The patient had reported a positive history due to allergy to fermented soy germs (nattō) confirmed by the finding of high levels of specific IgE antibodies; moreover, the patient had frequently been stung by jellyfish as a surfer from the age of 20 years. The genesis of anaphylaxis, as reported by this publication, is described after ingestion of fermented soy germs (rich of PGA) in patients already sensitized to PGA through the skin following jellyfish sting. Two publications [19,33] state that surfers are more likely to develop adverse reactions after ingesting jellyfish because they are more easily exposed to contact with them. This observation could alert people who practice water sports and have previously came into contact with jellyfish to avoid consuming them.

While in the article written by Inomata et al. [33], it seems to be a probable mechanism of cross-reactivity between soybean seeds and jellyfish, no cross-reactivity between jellyfish and soy beans (nattō) was demonstrated in an article published by Wakiguchi et al. [35], suggesting that there are also other allergens which could cause anaphylactic reaction. Further studies are needed to evaluate this and other possible associations.

The articles related to chemical risk highlighted that a careful prior assessment is due to the location of capture and to the breeding of jellyfish. The phenomenon of bioaccumulation, a process through which toxic polluting substances accumulate inside an organism in concentrations higher than those found in the surrounding environment, is typical of the marine species here treated [17,28,29]. For this reason, it is essential to carry out a careful environmental analysis in order to search possible marine pollutants such as pesticides, hydrocarbons, and heavy metals, before marketing jellyfish. Therefore, the capture of jellyfish in the open sea, far from estuaries or urbanized areas, would seem the best choice to be made.

Among the studies on heavy metals, aluminum toxicity is well documented [38]. There are also relationships between aluminum and the occurrence of neurodegenerative disorder, metabolic bone disease, anemia and even genotoxic activity [39,40]. For example, aluminum accumulation in the brain can potentiate oxidative and inflammatory events, leading to tissue damage and playing a key role in the Alzheimer’s disease (AD) etiology [41,42].

Another observation concerns the manufacturing process of jellyfish and the failure to comply with the rules governing [43] the use of chemical additives in food products. The processing technology, due to the wide use of alum represents a possible health risk: a study carried out concerning the inorganic constituents of the processed jellyfish highlighted how the percentages of aluminum are higher in the finished products than in the raw ones [44].

Aluminum, in fact, is used as a food additive, in the form of aluminum sulphates (E 520–523) and sodium aluminum phosphate (E 541) [45]. The provisional tolerable weekly intake (PTWI) of 2 mg/kg bw is the limit fixed by the evaluation of the Joint FAO/WHO Expert Committee on Food Additives (JECFA) [46]. The estimated daily dietary exposure to aluminum in the general population, assessed in several European countries, varied from 0.2 to 1.5 mg/kg bw/week at the mean and was up to 2.3 mg/kg bw/week in highly exposed consumers. The tolerable weekly intake (TWI) of 1 mg/kg bw/week is therefore likely to be exceeded in a significant part of the European population [47]. In China, “Standards for uses of food additives” (GB 2760-2014), a maximum aluminum residual level of 100 mg/kg in dry weight is established. The standard for alum in salted jellyfish could be set at no more than 1.8% [43].

The difference between Europe and China regarding aluminum safety range as additive does not allow to define an unambiguous limit. Furthermore, a study showed that, during product preparation, parameters such as exposure processing time, temperature, and the amount of alum used influence the retention and therefore the total concentration of aluminum in jellyfish tissues [25]. According to another study conducted by Ma et al. [30], a very low concentration of aluminum in fish products was observed, while on the contrary, high values of this metal were found in jellyfish caught in the same area.

Another problem could be related to the packaging of these products often without the necessary identification label for food use. Considering the growing number of risks linked to the chemical composition of packaging commercialized in the Asian countries, this aspect should be carefully evaluated.

### Limitation of the Study

The 14 works included in this systematic review are methodologically different. Four articles related to the allergenic risk must be interpreted with caution because, being case reports, they refer only to a single case and not to large population samples [18,33,34]. Regarding the allergenic risk, the population samples used in a study conducted by Amaral [19] are small (twenty-five patients), and this could limit the generalization of the results. Only six articles about allergenic risks that satisfy our research criteria have been included. No physical risks associated with the consumption of jellyfish were found with the temporal filter used. Another limit is the exiguous quantity of studies (one) about microbiological risk [17]. As concerns the chemical risk, we have to consider that the study conducted by Ma et al. [30] refers to a Chinese city, Tianjin, where the policies governing the amount of aluminum food additives have not been respected and, consequently, it cannot be affirmed whether or not the results of this work are generalizable. Yang et al. [32] in their study report other three limitations. First, study participants in the dietary survey already knew they were involved in the study; second, the results for children may be biased by overestimation since the food records for children were filled by their parents; and third, to estimate the dietary exposure, it has been used the median of aluminum concentration and the mean of consumption. Other limitations have been found in an article written by Zhang et al. [31] in which only three days of dietary records data were used, while individual habits of personal consumption are averaged over a prolonged period of time. In addition, not all food consumed was investigated, which may underestimate the exposure level. Other two studies conducted by Epstein et al. and Muñoz-Vera et al. [28,29] report altered heavy metal levels caused by the polluted places where jellyfish were caught. This aspect could limit the generalization of the results, since it tends to overestimate the concentration levels of heavy metal in comparison to those found in jellyfish caught in the open sea.

## 5. Conclusions

In the near future, jellyfish could become a valid alternative foodstuff with good safety for human consumption, although the number of publications in the literature about the risks assessed is rather small. This novel food is a very sustainable food source in every dimension such as costs, nutritional values, environmental impact, availability, and production process. Moreover, the use of jellyfish as a large-scale food can be a choice and an opportunity, limiting the excessive proliferation of this aquatic species as a consequence of climate change in the marine ecosystem. Perhaps, the only real challenge consists in overcoming the common mistrust by European consumers of what is “strange” and untypical, which can be achieved stimulating a new awareness to novel food. In conclusion, considering the evidence on the critical issues about the "jellyfish food product" and the commercial power in western markets, further investigations are needed in order to confirm the observed results and enhance the current knowledge regarding the consumption and the safety of jellyfish.

## Figures and Tables

**Figure 1 foods-09-00935-f001:**
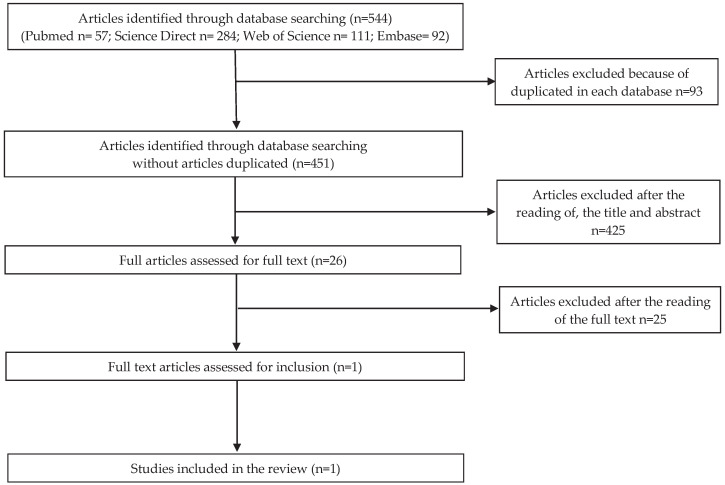
PRISMA (Preferred Reporting Items for Systematic reviews and Meta-Analyses) selection for microbiological risk.

**Figure 2 foods-09-00935-f002:**
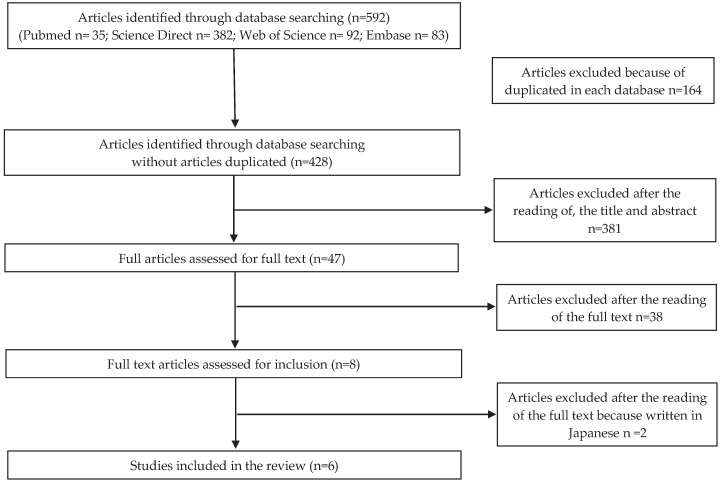
PRISMA (Preferred Reporting Items for Systematic reviews and Meta-Analyses) selection for allergenic risk.

**Figure 3 foods-09-00935-f003:**
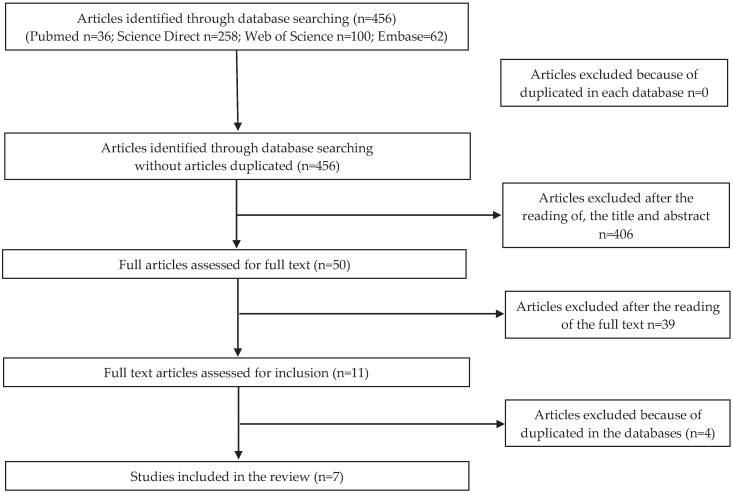
PRISMA (Preferred Reporting Items for Systematic reviews and Meta-Analyses) selection for chemical risk.

**Table 1 foods-09-00935-t001:** Main characteristics of the studies included in the systematic review.

Author, Year, and Country	Title	Aim	Type of Risk	Main Result
Raposo, A. et al. 2018 (Portugal)	Eating jellyfish: safety, chemical, and sensory properties.	The study evaluated microbiological, heavy metal, and allergenic control, and made sensory evaluation (using a sensory profile and acceptance test) to encourage the consumption of edible jellyfish in Western countries.	Microbiological	The microbiological analysis focused on pathogenic microorganism markers: *Aeromonas hydrophila*, *Listeria monocytogenes*, *Salmonella* spp., and *Vibrio* spp. The microbiological analysis was negative for all the tested pathogenic markers.
Raposo, A. et al. 2018 (Portugal)	Eating jellyfish: safety, chemical, and sensory properties.	The study evaluated microbiological, heavy metal, and allergenic control, and made sensory evaluation (using a sensory profile and acceptance test) to encourage the consumption of edible jellyfish in Western countries.	Allergenic	Twenty subjects, all allergic to fish, crustaceans and/or cephalopods, agreed to formally take part in the trial. They all had a previous history of severe systemic reactions, including 12 with anaphylaxis. Ten had asthma, 18 rhinitis, and 15 were sensitized to house dust mites. The trial consisted of a skin prick-to-prick test of the homogenized crude umbrella of *Catostylus tagi*. All 20 patients presented negative reactions to the SPPT with the raw umbrella of *Catostylus tagi*.
Li, Z. et al. 2017 (China)	Allergic shock caused by ingestion of cooked jellyfish: A case report.	The study assessed the relationship between allergic reactions after consuming jellyfish and a previous contact with jellyfish stings.	Allergenic	A history of jellyfish contact or sting might be an important allergic factor for individuals who consume any kind of jellyfish, within at least 1 year after being stung by a jellyfish.
Inomata, N. et al. 2014 (Japan)	Anaphylaxis caused by ingesting jellyfish in a subject with fermented soybean allergy: Possibility of epicutaneous sensitization to poly–gamma–glutamic acid by jellyfish stings.	The study assessed the causes of anaphylaxis reactions after jellyfish consumption in a subject allergic to fermented soybeans (natto).	Allergenic	The patient might have been sensitized to jellyfish nematocyte PGA by stings via the skin during surfing and consequently developed the anaphylactic reactions to ingested jellyfish and natto.
Wakiguchi, H. et al. 2018 (Japan)	*Lobonemoides robustus Stiasny* (jellyfish) anaphylaxis without poly–γ–glutamic acid sensitization.	The study assessed the relationship between consumption of soybeans (natto) and jellyfish stings and risk of developing anaphylaxis after jellyfish ingestion.	Allergenic	The results of the study suggest that the ingestion of *Lobonemoides robustus* can lead to anaphylaxis without any history of jellyfish sting. It has been administered an oral food challenge test to the patient for this species of jellyfish. The patient experienced the symptoms forty-five minutes after having ingested the jellyfish. The patient was diagnosed with anaphylaxis due to jellyfish ingestion but not with PGA; no cross-reactivity between jellyfish and natto was demonstrated.
Amaral, L. et al. 2018 (Portugal)	Jellyfish ingestion was safe for patients with crustaceans, cephalopods, and fish allergy.	The study established the safety of jellyfish ingestion in seafood allergic patients and to evaluate the willing to introduce jellyfish in their diet.	Allergenic	This study shows that *Catostylus tagi* umbrella may be safely consumed by seafood allergic individuals. All the patients with severe crustacean, cephalopods, and/or fish allergy, including 60% with anaphylaxis, tolerated all the pastes.
Okubo, Y. et al. 2015 (Japan)	Anaphylactic shock after the ingestion of jellyfish without a history of jellyfish contact or sting.	The study analyzed risk factors which occurred after ingestion of jellyfish without a clinal story of jellyfish sting in a 14 years old boy.	Allergenic	The patient had no history of jellyfish contact or stings. His sensitization to jellyfish may have been caused by a different mechanism from jellyfish sting. In this study, cross-reactivity and specific IgE antibodies against tropomyosin and other types of seafood have been investigated, which were all negative. It is possible that an unknown antigen induced the sensitization to the jellyfish or that the patient had ingested jellyfish long before this anaphylactic event without knowing, which caused the sensitization. Clinicians should be cautious of the possibility of anaphylaxis caused by jellyfish ingestion, with or without a history of jellyfish contact or sting.
Raposo, A. et al. 2018 (Portugal)	Eating jellyfish: safety, chemical, and sensory properties.	The study evaluated microbiological, heavy metal, and allergenic control, and made sensory evaluation (using a sensory profile and acceptance test) to encourage the consumption of edible jellyfish in Western countries.	Chemical	Of the 25 elements analyzed in the umbrella product, 11 (Al, B, C, Fe, H, K, Mg, Mn, N, Na, and P) showed a significant mass change, *p* < 0.1, compared to the total solids of the raw umbrella. Eight elements (Ca, Cd, Cr, Cu, I, Ni, S, and Zn) showed no significant differences. Particularly, Al was lower in cooked product than in raw; As, Co, Hg, Mo, Pb, Se, and V were not detected after cooking.
Ma, J. et al. 2019 (China)	A longitudinal assessment of aluminum contents in foodstuffs and aluminum intake of residents in Tianjin metropolis.	The study analyzed aluminum contents in foodstuffs over a 6-year span between 2010 and 2015 to assess the risk of dietary aluminum exposure in residents of Tianjin metropolis.	Chemical	Totally, 21.14% of food samples exceeded the recommended aluminum residue limit over the study period (100 mg/Kg). The lowest mean aluminum levels in food were detected in 2010, and the highest levels were found in 2015. The highest levels were found in jellyfish (433.28 ± 402.11 mg/kg), while the lowest aluminum levels were found in the other aquatic animal food products (2.26 ± 5.58 mg/kg).
Muñoz-Vera, A. et al. 2016 (Spain)	Patterns of trace element bioaccumulation in jellyfish *Rhizostoma pulmo* (*Cnidaria*, *Scyphozoa*) in a Mediterranean coastal lagoon from SE Spain.	The study analyzed the ability of *Rhizostoma pulmo* to bioaccumulate trace elements.	Chemical	High accumulative capacity has been seen in *Rhizostoma pulmo*, for the elements within the following ranges (dry weight, ppm): Al (0.74–66.14), Ti (0.27–2.71), Cr (0.24–25.56), Mn (0.20–12.15), Fe (4.77–398.49), Ni (0.21–119.52), Cu (0.11–11.88), Zn (10.07–82.86), As (3.57–130.31), Cd (0.00–0.15), Sn (22.52–87.97), and Pb (0.07–6.04). Except for Sn where the lowest and highest values were obtained in the oral arms; for other elements, the lowest values were obtained for the bell tissues, while the highest values always corresponded to oral arm tissues.
Epstein, H.E. et al. 2016 (Australia)	Fine-scale detection of pollutants by a benthic marine jellyfish.	The study examined the uptake and retention rates of the trace metals copper (Cu) and zinc (Zn) in *C. maremetens* tissue in the presence and absence of multiple stressors.	Chemical	Metal accumulation in jellyfish tissue began rapidly within 24 h of exposure to treated water. Cu concentrations were significantly higher under high nutrient conditions (ANOVA: F_1,16_ = 7.436, *p* = 0.015), reaching 2.627 ± 0.031 µg/g, an increase of approximately 18.1% from ambient concentrations. Zn concentration reached 4.751 ± 0.008 µg/g in ambient conditions by day six and was 17.2% higher under high nutrient conditions, reaching 5.738 ± 0.012 µg/g, but was not statistically significant.
Zhang, H., et al. 2015 (China)	Aluminum in food and daily dietary intake assessment from 15 food groups in Zhejiang province, China.	The study evaluated the dietary Al intake level and health risk in residents of Hejiang Province, China.	Chemical	Dietary exposure to aluminum was estimated for Zhejiang residents. High aluminum levels were found in jellyfish (mean 4862 mg/kg), laver (mean 455.2 mg/kg) and fried twisted cruller (mean 392.4 mg/kg). The estimated average aluminum intake via 15 food groups is 1.15 mg/kg bw/week, which is currently regarded as lower compared with the provisional tolerable weekly intake (PTWI) of 2 mg/kg bw/week. Jellyfish contains the highest Al concentration and is the main Al contributor, providing 37.63% of the Al intake.
Khong, N. et al. 2015 (Malaysia)	Nutritional composition and total collagen content of three commercially important edible jellyfish.	The study evaluated the potential of three commercially significant edible jellyfish species (*Acromitus hardenbergi*, *Rhopilema hispidum,* and *Rhopilema esculentum*) as nutritional and functional food ingredients.	Chemical	Hazardous elements such as lead (Pb), cadmium (Cd), mercury (Hg), and inorganic tin (Sn) were not within the detection limit (0.01 weight %). Elements in jellyfish especially trace elements are highly affected by the habitats of the jellyfish population. The jellyfish were found to contain a high amount of water, whereas the dry mass was rich in protein (20.0–53.9 g/100g D.W.) and minerals (15.9–57.2 g/100g D.W.), while low in fats (1.0–4.9 kcal/g D.W.) and calories (1.0–4.9 kcal/g D.W.) Collagen was found to be the major protein in edible jellyfish (122.64–693.92 30 mg/g D.W.), and glycine, proline, and glutamic acid were found to be the dominant amino acids in edible jellyfish.
Yang, M., et al. 2014 (China)	Dietary Exposure to Aluminum and Health Risk Assessment in the Residents of Shenzhen, China.	The study evaluated the dietary aluminum intake level in residents of Shenzhen, China.	Chemical	A total of 1399 food samples were analyzed for aluminum concentration. Aluminum concentrations of 176 samples were lower than the limit of detection (LOD). Aluminum concentrations were varied considerably among the food samples (ranged from not detected to 1250.037 mg/kg). Among all 17 food groups, jellyfish has the highest aluminum concentration, ranging from 318.334 to 1000.359 mg/kg with a median of 527.500 mg/kg. The highest aluminum intake concerned children with an exposure level of 3.356 mg/kg bw/week and 3.248 mg/kg bw/week in 0–2 and 3–13 age groups, respectively, which is higher than the allowed threshold.

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
