# Peer review of "A Systematic Review of Risk Assessment Associated with Jellyfish Consumption as a Potential Novel Food"

_foods, 2020, doi:10.3390/foods9070935_

Round 1
Reviewer 1 Report
The manuscript titled “Novel food: a systematic review of risk assessment associated with jellyfish consumption” presents a valuable summary and compilation of a very scarce topic in the literature regarding the risks associated to the consumption of jellyfish, especially in a background of food depletion and the need of finding new food sources. The article is well written and presents a logical structure, and the review was conducted in a systematic form. There are minor revisions that should be conducted by the authors to be suitable for publication in Foods.
- TITLE: Although the title reflects the review’s content, I do not consider that the expression “Novel food” matches within the title. I would suggest: “A systematic review of risk assessment associated with jellyfish consumption as a potential novel food.”
- ABSTRACT: As this was a systematic review, the abstract should also include results about the overall number of documents that were found, or the number of documents that were covered for each risk. Besides, it is mentioned “environmental sustainability”, but there was little or no information in the reviews pointing to this topic, so I would suggest to delete it.
INTRODUCTION
- Line 35: I feel this reference is very old. Please consider changing it for a newer one.
- Line 45: Although deforestation is a cause for loss of plant biodiversity, it is not the leading cause. I would suggest adding more direct causes such as its replacement by more productive or commercially attractive crop varieties or the use of fertilizers of chemical products to increase the production of certain crops without considering the local varieties.
RESULTS
- Line 151: Please use italics for all the scientific names of species such as Catostylus tagi, Aeromonas hydrophila, Listeria monocytogenes, etc.
- Line 167: Please use a standard way of presenting units through the manuscript. For instance, mg/g.
- Line 169: Please change “ability” in the expression: “(...) assessed the ability of a jellyfish species (Rhizostoma pulmo) to bioaccumulate...” because bioaccumulation is not an intended activity of animals, but accidental.
- Line 216: Was it: “Had not been stung severely by...”
- Line 239: Please change “a” to a capital letter in “a prick-to-prick test...” The same for “(...) the patient was diagnosed...” (Line 240).
- Table 1: I do not feel you should start each line from the “Aim” column as “the study aimed to”. If the column title is “Aim”, then the text can avoid this term.
DISCUSSION
- Line 313: Please spells “PTWI”.
Reviewer 2 Report
The article ‘Novel food: A systematic review of risk assessment associated with jellyfish consumption’ is an interesting review assessing microbiological, chemical, physical and allergenic risks associated with jellyfish consumption.
More information and references about the nutritional interest of jellyfish apart from long chain fatty acids, antioxidants, collagen, and taurine, should be included in the manuscript (in the Introduction section). Also the organoleptic characteristics of the jellyfish should have been included in the introduction.
Jellyfish is an unknown food in Western countries and it is important to highlight main strengths to show the advantage of its consumption as well as for sustainability reasons.
The increase in the global population will also lead to an increase in protein consumption especially in developing and underdeveloped countries. Thus, the demand on protein-rich foods is expected to increase in the next decades, so the content in protein of the proposed novel food sources such as jellyfish should be considered in the manuscript.
Probably due to jellyfish is not considered a food in Western countries, a low number of articles especially regarding allergens are in the literature and have been reported in this study.
The text shoud be unifixed to Amerian English or British English. It is mostly written in American, then words such as 'sensitised' or 'homogenised' in Table 1 should be changed. Also 'aluminum' should be used instead of 'aluminium'.
Round 2
Reviewer 2 Report
The document has been corrected according to the suggestions and the contents have improved significantly.